# Methionine Metabolism Is Down-Regulated in Heart of Long-Lived Mammals

**DOI:** 10.3390/biology11121821

**Published:** 2022-12-14

**Authors:** Natalia Mota-Martorell, Mariona Jové, Rebeca Berdún, Èlia Òbis, Gustavo Barja, Reinald Pamplona

**Affiliations:** 1Department of Experimental Medicine, University of Lleida-Biomedical Research Institute of Lleida (UdL-IRBLleida), 25008 Lleida, Spain; 2Department of Genetics, Physiology and Microbiology, Complutense University, 28040 Madrid, Spain

**Keywords:** amino acids, methionine, longevity, mass spectrometry, transmethylation pathway, transsulfuration pathway

## Abstract

**Simple Summary:**

Long-lived species have evolved by reducing the rate of aging, which is an inherent consequence of oxidative metabolism. Hence, species that live longer benefit from more efficient intracellular metabolic pathways, including lipid, protein, and carbohydrate metabolism. The aim of this work is to determine whether the content of proteins’ building blocks, named amino acids, are related with mammalian longevity. This was accomplished by analyzing the amino acid content in the hearts of seven mammalian species with a longevity ranging from 3.8 to 57 years. Our findings demonstrate that the heart’s content of amino acids differs between species and is globally lower in long-lived species. Moreover, long-lived species have lower content of amino acids containing sulfur, such as methionine and its related metabolites. Our results support the existence of metabolic adaptations in terms of sulfur-containing amino acids. As has been described previously, our work supports the idea that the human population could benefit from reduced calorie intake, which would lead to reduced age-related diseases and healthier aging.

**Abstract:**

Methionine constitutes a central hub of intracellular metabolic adaptations leading to an extended longevity (maximum lifespan). The present study follows a comparative approach analyzing methionine and related metabolite and amino acid profiles using an LC-MS/MS platform in the hearts of seven mammalian species with a longevity ranging from 3.8 to 57 years. Our findings demonstrate the existence of species-specific heart phenotypes associated with high longevity characterized by: (i) low concentration of methionine and its related sulphur-containing metabolites; (ii) low amino acid pool; and (iii) low choline concentration. Our results support the existence of heart metabotypes characterized by a down-regulation in long-lived species, supporting the idea that in longevity, less is more.

## 1. Introduction

Known adaptations leading to long-lived mammalian species phenotypes include a low rate of mitochondrial ROS production (mitROSp), subsequently low oxidative damage to mitDNA, and the presence of macromolecules, like cell membrane lipids, which are highly resistant to oxidative modification [1,2,3]. Among the multiple mitROS generators, the ROS production at complex I (Cx I), linked to aging [1,4,5], was recently localized at the FeS cluster N1a inside the NDUFV2 subunit of the hydrophilic domain [4]. Both quantitative [4,6] and qualitative [7] changes seem to be mechanistically involved in the evolution of the low mitROSp rate of long-lived species.

Macromolecular longevity adaptations are genetically determined and regulated by specific signaling pathways [8], including the mechanistic target of the rapamycin (mTOR) pathway [9], afferently targeting the nuclear aging program [8]. Differential target gene expression patterns arising from this program [8,10] finally produce specific transcriptomic, proteomic, metabolomic, and lipidomic profiles [11,12,13], including macromolecular adaptations concerning nucleotides in DNA, amino acids in proteins, and lipid profiles in cell membranes [1,3,14,15].

Proteins evolved to become more oxidation-resistant by regulating their sulfur-containing residue content, such as methionine and cysteine, as an expression of the redox conditions of the animal species. Thus, adaptative decreases in methionine [16,17] and cysteine protein content [18], as well as in free methionine tissue concentration [9,19], have evolved as adaptations of long-lived animal species. Beyond their roles in protein structure and function, methionine and cysteine are also involved in a complex metabolic network (globally referred to as methionine metabolism) that can be divided into the methionine cycle or transmethylation pathway, the transsulfuration pathway, and the methionine salvage pathway, including polyamine biosynthesis [20,21]. Significantly, available evidence points to methionine metabolism and its specific modulation as another feasible determinant of animal longevity [9,19,22].

The specific aim of this study is to investigate at the tissue level the global methionine metabolic phenotype of long-lived mammalian species. In this investigation, we have unambiguously detected and quantified 31 different molecules in the heart of seven mammalian species with different longevities varying from 3.8 years in rats to 57 years in horses. The metabolites detected and quantified were: (a) methionine and its related metabolites, including the intermediates of the transmethylation pathway S-adenosylmethionine (SAM), S-adenosylhomocysteine (SAH), and homocysteine; betaine and spermidine as metabolites involved in the regeneration of methionine levels in the remethylation and methionine salvage pathways, respectively, acting as substrates and intermediates; the intermediates of the transsulfuration pathway cystathionine and cysteine; taurine and glutathione (GSH) as final products of the transsulfuration pathway; and the vitamin B6 metabolites acting as transsulfuration enzyme cofactors pyridoxal, pyridoxal 5′ phosphate (PLP), and pyridoxamine; (b) additional amino acids including eight non-polar amino acids (alanine, glycine, isoleucine, leucine, phenylalanine, proline, tryptophan, and valine), four polar uncharged amino acids (asparagine, serine, threonine, and tyrosine), two polar negatively charged amino acids (aspartate and glutamate), and two polar positively charged amino acids (arginine and histidine); and (c) the methionine and SAM derived lipid intermediates choline and carnitine. The species-specific longevity-associated heart metabolite phenotypic profiles were determined using an LC-MS/MS platform.

## 2. Materials and Methods

### 2.1. Chemicals

Unless otherwise specified, all reagents were from Sigma-Aldrich (Sigma, Madrid, Spain) and of the highest purity available.

### 2.2. Samples

Mammalian species included in the study were male adult specimens with an age representing 15–30% of their longevity. The recorded values for longevity (in years), according to the AnAge website (https://genomics.senescence.info/species/query.php?search (accessed on 30 June 2020)), were: rat (*Rattus norvegicus*, n = 5), 3.8; mouse (*Mus musculus*, n = 4), 4; rabbit (*Oryctolagus cuniculus*, n = 5), 9; guinea pig (*Cavia porcellus*, n = 5)*,* 12; cow (*Bos taurus*, n = 4), 20; pig (*Sus scrofa*, n = 5), 27; and horse (*Equus caballus*, n = 5), 57. Rodents and rabbits were obtained from rodent husbandries and sacrificed by decapitation, whereas pigs, cows, and horses were raised at farms and sacrificed at the abattoir where samples were quickly obtained after death. Heart samples were taken from ventricles of 4–5 animals of each species within 5 min after death, were immediately frozen in liquid nitrogen, and were stored within the next two hours at −80 °C, remaining at that temperature until analyses.

### 2.3. Sample Homogeneization and Quantification

Heart tissue (≈50 mg of whole tissue) was homogenized in a buffer containing 180 mM KCl, 5 mM MOPS, 2 mM EDTA, 1 mM DTPAC adjusted to pH = 7.4. Prior to homogenization, 1 μM BHT and a mix of protease (GE80-6501-23, Sigma, Madrid, Spain) and phosphatase (1 mM Na_3_VO_4_, 1M NaF) inhibitors were added. After a brief centrifugation (1000 rpm at 4 °C for 3 min), the protein concentration was measured in the supernatants using the Bradford method (500-0006, Bio-Rad Laboratories, Barcelona, Spain). Prior to analysis, protein concentration was normalized in all samples.

### 2.4. Sample Processing

Heart metabolite extraction was performed based on a previously described method [23]. Briefly, 10 µL of sample homogenates were added to 30 µL of cold methanol containing 1 μg/mL of Phe-^13^C as an internal standard and 1 μM BHT as an antioxidant. Then, samples were incubated at room temperature for 15 min and centrifuged at 12.000 *g* for 3 min. Finally, the supernatants were filtrated through a 0.22 μm diameter organic filter (CLS8169, Sigma, Madrid, Spain) and were transferred to vials with glass inserts (Agilent Technologies, Barcelona, Spain) for further analysis.

Sulphur-containing metabolites were extracted using a previously described method [24]. Briefly, 2 µL of 5% DTT diluted in methanol (*m*/*v*) were added to 10 µL of sample homogenates. The resulting solution was vortexed for 1 min and allowed to stand at room temperature for 10 min. For protein precipitation, 40 µL of acetonitrile containing 0.1% formic acid (*v*/*v*), 0.05% trifluoroacetic acid (*v*/*v*), and 1 µg/mL of Phe-^13^C as an internal standard were added to the sample, and the solution was vortexed for 2 min. Then, samples were incubated at room temperature for 15 min and centrifuged at 12.000× *g* for 3 min. Finally, the supernatants were filtrated through a 0.22 μm diameter organic filter (CLS8169, Sigma, Madrid, Spain) and were transferred to vials with glass inserts (Agilent Technologies, Barcelona, Spain) for further analysis.

### 2.5. Analysis Conditions

The individual conditions for the detection and quantification of heart metabolites are listed in Table A1 (Appendix A). For non-sulphur-containing metabolites, 2 µL of extracted sample was injected based on the method described [23]. Chromatographic separation was achieved on a reversed-phase column (Zorbax SB-Aq 2.1 × 50 mm, 1.8 µm particle size, Agilent Technologies, Barcelona, Spain) equipped with a pre-column (Zorba-SB-C8 Rapid Resolution Cartridge, 2.1 × 30 mm, 3.5 µm particle size, Agilent Technologies, Barcelona, Spain) with a column temperature of 60 °C. The flow rate was 0.6 mL/min for 19 min. Solvent A was composed of water containing 0.2% acetic acid (*v*/*v*), and solvent B was composed of methanol containing 0.2% acetic acid (*v*/*v*). The gradient started at 2% of solvent B and increased to 98% B in 13 min and held for 6 min. Post-time was established at 5 min. Electrospray ionization was performed in both positive and negative ion mode (depending on the target metabolite) using N_2_ at a pressure of 50 psi for the nebulizer with a flow of 12 L/min and a temperature of 325 °C, respectively.

For sulphur-containing metabolites, 10 µL of extracted sample was injected based on the described method [24]. Chromatographic separation was achieved on a reversed-phase Supelcosil LC-CN column (Supelco of 4.6 × 250 mm, 5 µm particle size, Sigma, Madrid, Spain) with a column temperature of 30 °C. The flow rate was maintained at 0.5 mL/min for 10 min using a mobile phase of 10:90 acetonitrile/water with 0.1% formic acid (*v*/*v*). Electrospray ionization was performed in both positive and negative ion mode (depending on the target metabolite) using N_2_ at a pressure of 50 psi for the nebulizer with a flow of 12 L/min and a temperature of 325 °C, respectively.

Data were collected using the MassHunter Data Analysis Software (Agilent Technologies, Santa Clara, CA, USA). Samples were decoded and randomized before injection. Metabolite extraction quality controls (heart samples with internal Phe-^13^C) were injected every 10 samples. Peak determination and peak area integration were carried out with MassHunter Quantitative Analyses (Agilent Technologies, Santa Clara, CA, USA). Free amino acids and metabolite content are expressed as mmol per heart mass (g).

### 2.6. Equipment

The analysis was performed through liquid chromatography coupled to a hybrid mass spectrometer with electrospray ionization and a triple quadrupole mass analyzer. The liquid chromatography system was an ultra-performance liquid chromatography model 1290 coupled to LC-ESI-QqQ-MS/MS model 6420, both from Agilent Technologies (Barcelona, Spain).

### 2.7. Statistics

Prior to statistical analyses, data were pre-treated (auto-scaled and log-transformed). Multivariate statistics were performed using Metaboanalyst software [25], including principal component analysis (PCA), partial-least squares discriminant-analysis (PLS-DA), and hierarchical clustering analysis represented by a heat map. Pearson correlation was performed using IBM SPSS Statistics (v24.0.0.0). Linear regression was performed and plotted using GraphPad prism (v8.0.1). Phylogenetic generalized least squares (PGLS) regression was performed and plotted using RStudio (v1.1.453). Functions used were included in the packages caper [26] and ggplot2 [27]. The phylogenetic tree was constructed using taxa names as described previously [28].

## 3. Results

### 3.1. Multivariate Statistics Reveal a Heart-Species-Specific Methionine-Related Metabolite Profile

In order to determine whether the concentration of heart methionine and its related metabolites differed among mammals, multivariate statistics were applied. Non-supervised PCA suggested the existence of a species-specific methionine-related metabolite profile in the hearts (Figure 1A), capable of explaining 70.9% of sample variability. These results were confirmed after performing a hierarchical clustering of the samples represented by a heat map, which revealed a good sample clusterization of individual specimens according to its species (Figure 1B). Furthermore, average specimen values suggested the existence of different methionine profiles for short-lived species like rodents (mouse, rat, and guinea pig) and rabbit, compared with longer-living species such as pig, cow, and horse (Figure 1C). The PLS-DA model reinforced the discrimination power of these metabolites (R^2^ = 0.6, Q^2^ = 0.5 and accuracy = 0.6, *p* < 0.001), having the most impact for clustering taurine, spermidine, PLP, cystathionine, and pyridoxal (Figure 1D).

### 3.2. Low Levels of Methionine Related Metabolites in the Heart of Long-Lived Animals

The species-specific differences in content of heart methionine and its related metabolites were evaluated as a function of species longevity. Globally, the concentration of transmethylation and transsulfuration metabolites negatively correlated with species longevity (Figure 2A, Table A1 and Table A2). The transmethylation intermediates methionine and SAM, as well as spermidine, were lower in longer living than in short-lived species. Contrary, the levels of SAH positively correlate with animal longevity. The concentrations of the transsulfuration intermediate cystathionine, as well as that of taurine (a sulphur-containing metabolite synthetized from cysteine), were also lower in long-lived species. Accordingly, the vitamin B6 intermediates pyridoxal and PLP (cofactors of transsulfuration enzymes) were also negatively correlated with species longevity (Figure 2B). No changes in the heart content of the transmethylation intermediates homocysteine and betaine, as well as on levels of the transsulfuration metabolites cysteine and glutathione, were observed as a function of longevity. Altogether, these results suggest that a global decrease in methionine metabolism and sulphur-containing metabolites has occurred during the evolution of long-lived mammals.

### 3.3. Amino Acid Content Is Decreased in Heart from Long-Lived Animals

Since heart methionine metabolism is associated with animal longevity, we hypothesized that the content of other amino acids could also express an adaptation to longevity. In order to test this, we have unambiguously detected 16 additional amino acids different from cysteine and methionine. Multivariate statistics was applied to determine whether the heart amino acid contents can define animal species. Non-supervised PCA suggested the existence of a different heart amino acid profile in different animal species (Figure 3A), able to explain 76.4% of sample variability. A hierarchical clustering of the samples represented by a heat map revealed a good sample clusterization for specimens of a selected species according to the relative amino acid heart abundance (Figure 3B). Average animal species values suggested, again, the existence of a different amino acids heart profile for short-living species such as rodents (mouse, rat and guinea pig), and for longer living species such as pig, cow and horse (Figure 3C). Among them, the threonine and aspartate arose as the most important metabolites defining the first component of PLS-DA model (R^2^ = 0.5, Q^2^ = 0.3 and accuracy = 0.5, *p* < 0.001) (Figure 3D).

The specific changes in heart amino acid content across animal longevity were also evaluated. Among the 16 detected amino acids (methionine and cysteine amino acids were not included), we observed a generally low concentration for most heart amino acids in long-lived animals (Figure 4, Table A1 and Table A2). We found a statistically significantly negative correlation with longevity for the concentration of non-polar amino acids such as leucine, isoleucine, phenylalanine, tryptophan, glycine, proline, and valine; for the levels of polar uncharged amino acids asparagine, threonine, and tyrosine; and for the levels of polar charged amino acids arginine, histidine, aspartate, and glutamate. Alanine heart content was the only amino acid which was positively correlated with species longevity. Serine heart content was the only amino acid that did not show a statistically significant correlation with mammalian longevity.

### 3.4. Heart Metabolome Is Also Related to Longevity concerning Specific Lipid Intermediates

Methionine metabolism participates in the biosynthesis of lipid intermediates such as choline and carnitine (Table A1 and Table A2). Although carnitine heart content did not show statistically significant differences across animal longevity, the heart choline concentration was negatively correlated with species longevity (Figure 5).

### 3.5. Methionine-Related Metabolites and Amino Acids Also Correlate with Longevity after Controlling for Phylogenetic Relationships

Animal species are evolutionarily related, and closely related species often have similar traits due to inheritance from a common ancestor. Most statistical analyses, like linear regression, assume data independence, which might not be necessarily true of data obtained from these phylogenetically related species. In order to find associations between longevity and heart methionine metabolites and amino acids, we applied a phylogenetic comparative correction method: phylogenetically generalized least squares (PGLS) regression. A phylogenetic tree evolutionarily relating the species in our study was inferred and constructed (Figure 6A).

First, under the assumption of a Brownian motion model of evolution (a branching, random walk of trait values from an ancestral value at the root to the tips of the tree [28]), we estimated the Pagel’s λ. This allowed us to measure the phylogenetic signal indicating the relative extent to which a traits’ correlation among close relatives matches a Brownian motion model of trait evolution. Pagel’s λ ranges from 0 to 1, where a λ = 1 value indicates that trait similarities between species are influenced by phylogenetic relationships, λ = 0 indicates that trait similarities between species are independent of phylogenetic relationships, and 0 < λ < 1 indicates different intermediate levels of phylogenetic signal. According to the estimated λ value, we classified the measured traits according to its association degree with phylogenetic relationships (Table A2): (i) independent (λ = 0), (ii) low dependence (λ < 0.6), (iii) mild dependence (λ > 0.6), and (iv) strong dependence (λ = 1), representing 63% of the analyzed metabolites. Finally, we applied a PGLS regression. This revealed that the heart content of methionine-related metabolites, such as methionine (r = −0.74, *p* = 0.007) and PLP (r = −0.67, *p* = 0.001), and that of the amino acids proline (r = −0.74, *p* = 0.019) and tyrosine (r = −0.66, *p* = 0.04) was negatively correlated with animal longevity also after controlling for phylogenetic relationships (Figure 6B, Table A2).

## 4. Discussion

Cell physiology and molecular composition are influenced by specific tightly regulated longevity-related gene expression patterns [29,30,31]. Diverse inter-species studies have reported the existence of species-specific tissue transcriptomics [10,32,33], proteomics [34], lipidomics [3,35], and metabolomics [13,19,36,37,38,39] profiles associated with animal longevity.

Methionine, an evolutionary adaptive redox-reactive residue with the capacity to protect cells from damage by ROS [40], is increasingly considered to be associated with longevity. Thus, lesser-encoded methionine in mitochondrial DNA [17] and decreased structural methionine content in tissue proteins [1] have been described in long-lived species. These findings are in line with the idea that the constitutive composition of cell macromolecular components and antioxidant systems are adapted in a species-specific way to the oxidative conditions, in particular to the mitROSp, and they are inversely correlated to animal longevity [1,5]. Thus, as in the case of endogenous cell antioxidants, constitutively less methionine protective residues would be needed in the proteins of long-lived animal species because they have low mitROS production rates.

Beyond its quantitative presence and meaning in protein structure and function, free methionine feeds into complex metabolic pathways, including the methionine cycle, the transsulfuration pathway, and polyamine biosynthesis. Significantly, different findings seem to indicate that the compositional observation at the protein level can be extended to the free methionine form. Thus, available evidence related to free methionine and longevity demonstrates lower concentrations of free plasma methionine in long-lived species [23] and exceptionally longevous animal species like the naked mole rat [19,36]. In this context, our study reported the existence of species-specific tissue (heart) profiles and association with longevity for metabolites belonging to methionine metabolism. More specifically, our data support the idea of an evolutionary metabolic adaptation targeted to maintain a lower concentration of methionine and its related transmethylation and transsulfuration metabolites in the hearts of long-lived mammalian species. Reinforcing this concept, the low expression of cystathionine β-synthase (CBS), as well as adenosyl methionine decarboxylase and spermidine synthase in long-lived mammalian species, was recently demonstrated in a transcriptomic comparative study [33]. Consistent with our findings, a global decrease in methionine and related metabolites was also found in long-lived fly strains [41], worms [42], and mice [43].

As described previously, these metabolic adaptations might vary between tissues. After comparing the results obtained in heart and plasma [23], we observe that the longevity effect in methionine and SAM reduction is similar in both tissues. However, changes correlated with species longevity are stronger in plasma than in the hearts of species. Accordingly, reduced content of cysteine and glutathione or increased content of SAH, betaine, and choline are stronger in plasma than in the heart (for cysteine: −55% in heart and −86% in plasma; for glutathione: −16% in heart and −97% in plasma; for SAH: +104% in heart and +921% in plasma; for betaine: +180% in heart and +571% in plasma; for choline: +15% in heart and +165% in plasma). Nevertheless, the reduction in cystathionine, taurine, asparagine, glutamate, and threonine is stronger in the heart than in plasma (for cystathionine: −87% in heart and −72% in plasma; for taurine: −96% in heart and 76% in plasma; for asparagine: −91% in heart and −78% in plasma; for glutamate: −66% in heart and −32% in plasma; for threonine: −82% in heart and −56% in plasma). These differences might be attributed to specific needs of the heart. The resulting reduced content of methionine-related metabolites in the heart could be due to an improved anabolism or catabolism. Long-lived species have evolved producing less ROS and synthesizing intracellular structures that are resistant to oxidation, which might be understood as an improved anabolism. This results in a lower amount of oxidized and altered intracellular structures that are rapidly eliminated by inducing autophagy [9], which results in an improved catabolism. In terms of protein metabolism, the protein turnover rate is lower in long-lived species compared to short-lived ones [44]. A recently published study in flies tracking isotopically labelled methionine revealed that methionine flux is altered during aging. In the same study, authors found that the expression of *Methioninase*, an enzyme that degrades methionine, extended the lifespan and reduced methionine flux into the TSP independently of methionine intake [45]. However, it cannot be ruled out the possibility that this profile might result from a reduced import of these metabolites, which are mostly provided by the liver through the bloodstream. Interestingly, the expression of the methionine transporter SLC43A2 in tumor cells leads to increased intracellular content of methionine [46]. However, the effects of these transporters in longevity determination remains to be elucidated. Significantly, the low concentrations of these metabolites probably have consequences in the metabolic pathways where they participate, also inducing adaptations related to longevity. Effectively, methionine via SAM regulates the mTOR signaling pathway [47], and a lower amount of activity of the mTOR complex 1 (mTORC1) has been described in long-lived species [9]. Similarly, low methionine and derived metabolites can determine epigenetic changes related to longevity, and, in fact, low expression of methyl transferases (for instance, Dnmt1) and epigenetic modifications in long-lived species [33,48,49] have been observed. Interestingly, the production of hydrogen sulfide (H_2_S) through the TSP is essential for promoting health and the longevity extension effect under specific nutritional interventions such as dietary and methionine restriction [50,51]. The TSP activity and, consequently, H_2_S production also depend on mTORC1 activity [51]. Furthermore, low methionine concentration can condition a lower lipid biosynthesis, and, indeed, a lower phospholipid biosynthesis in long-lived mammalian species has also been described [3]. Reinforcing this last observation, our results show that heart choline concentration is negatively correlated with mammalian longevity, which is also in line with the low plasma choline content also observed in the naked mole rat [36]. Choline is the precursor of phosphocholines (PC), which can be metabolized into lysoPC or phosphoserine in the mitochondria by exchanging choline and serine groups. Since betaine is synthesized from choline, it can be considered as a lipid intermediate that connects methionine and mitochondrial metabolism. Hence, a low concentration of free methionine and related metabolites is associated with an attenuation of different signaling and metabolic pathways which are an inherent characteristic of long-lived animal species. Consistent with our observations, methionine restriction as a nutritional intervention extends animal longevity through the attenuation of oxidative conditions and down-regulation of signaling pathways and their downstream effects [5]. However, more studies evaluating the metabolic flux should be performed to evaluate these processes.

## 5. Conclusions

In agreement with the idea that less is more, free amino acids were also lower in the hearts of long-lived compared to short-lived animal species, which is in line with previous studies performed in long-lived worms [52] and the plasma of naked mole rats [19,36]. Probably, this phenotype in the heart expresses the strong regulation of protein translation aimed at ensuring the proper fidelity described in long-lived species [53]. Accordingly, this might result from molecular mechanisms described in long-lived species, such as a low rate of protein synthesis via mTOR [9] or reduced protein turnover [44]. Notably, alanine was an exception, since its content was increased in the hearts of long-lived mammals. Interestingly, free alanine enters the Cahill cycle or glucose–alanine cycle, becoming an important energy fuel that can be metabolized into pyruvate in the liver [54]. The breakdown of carbohydrates and fatty acids supplies up to 90% of heart energy [55]. Although the biological relevance of having an increased heart alanine content in terms of longevity remains to be elucidated, we postulate that this might be associated to the maintenance of proper energy levels.

One limitation of this study is that it only provides a broad picture of the steady-state levels of the various metabolites and does not measure the metabolic flux. Moreover, our study does not prove causality between the metabolites and longevity traits, although different experimental studies demonstrate that changes in methionine metabolism metabolite levels may influence longevity. Furthermore, the number of metabolites quantified here only represents a fraction of the whole metabolome, and potentially relevant candidates may have been missed by the targeted approach applied. However, our work describes a heart species-specific longevity profile associated with a global adaptation of methionine metabolism, suggesting several new hypotheses to be tested.

## Figures and Tables

**Figure 1 biology-11-01821-f001:**
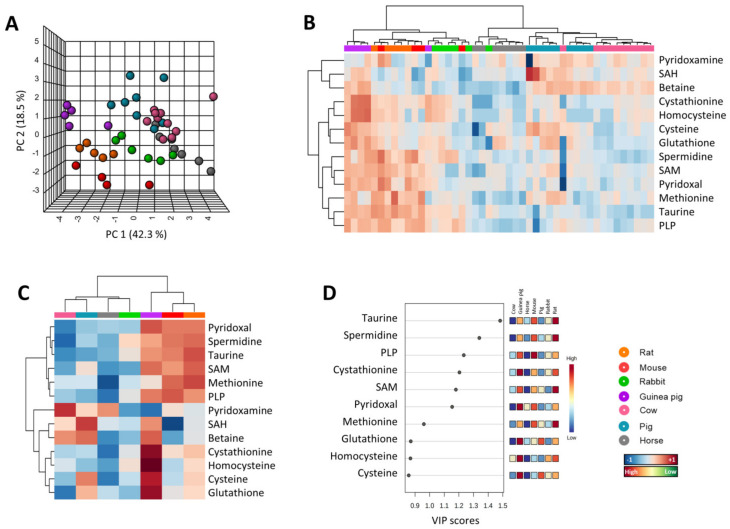
Multivariate statistics reveal a specific methionine heart profile for mammals. (**A**) Principal component analyses (PCA) representation of methionine-related metabolites. X: Principal component 1 (PC1); Y: Principal component 2 (PC2); Z: Principal component 3 (PC3). PC3 = 10.1% (not shown). (**B**) Hierarchical clustering of individual animal samples according to metabolite abundance. (**C**) Hierarchical clustering of animal species according to average metabolite abundance. (**D**) Variable importance projection (VIP) scores indicating the elements which contribute most to defining the first component of a partial least squares discriminant analysis (PLS-DA).

**Figure 2 biology-11-01821-f002:**
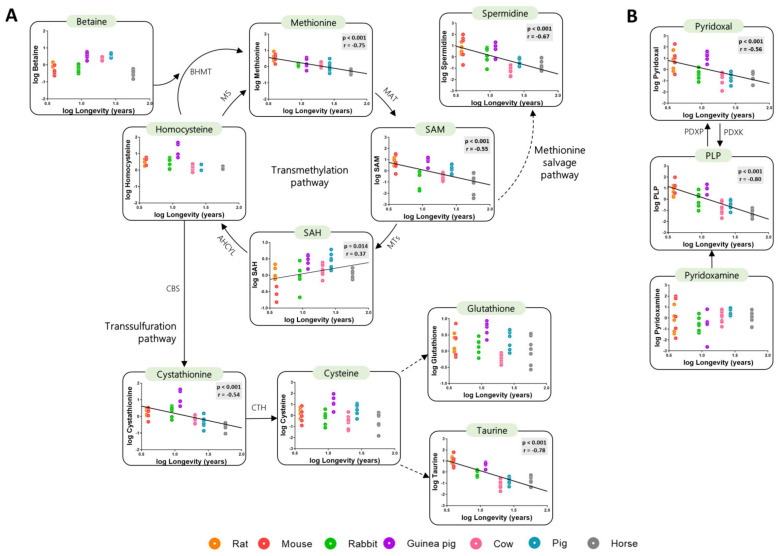
Heart methionine-related metabolites are negatively correlated with animal longevity. Heart concentration of metabolites involved in the transmethylation and transsulfuration pathways (**A**) and vitamin B6 metabolites (**B**). The concentrations of heart metabolites are reported in log(mmol/g tissue). Pearson correlation was performed. Linear regression was applied when significant relationships were found. The minimum significance level was set at *p* < 0.05. All metabolite values were log-transformed in order to accomplish the assumptions of normality. Enzyme codes refer to: Methionine synthase (MS); Betaine-Homocysteine S-methyltransferase (BHMT); Methionine adenosyltransferase (MAT); Methyltransferases (MTs); Adenosylhomocysteinase-like 1 (AHCYL); Cystathionine-β-synthase (CBS); Cystathionine-γ-lyase (CTH); Pyridoxal kinase (PDXK); Pyridoxal phosphatase (PDXP).

**Figure 3 biology-11-01821-f003:**
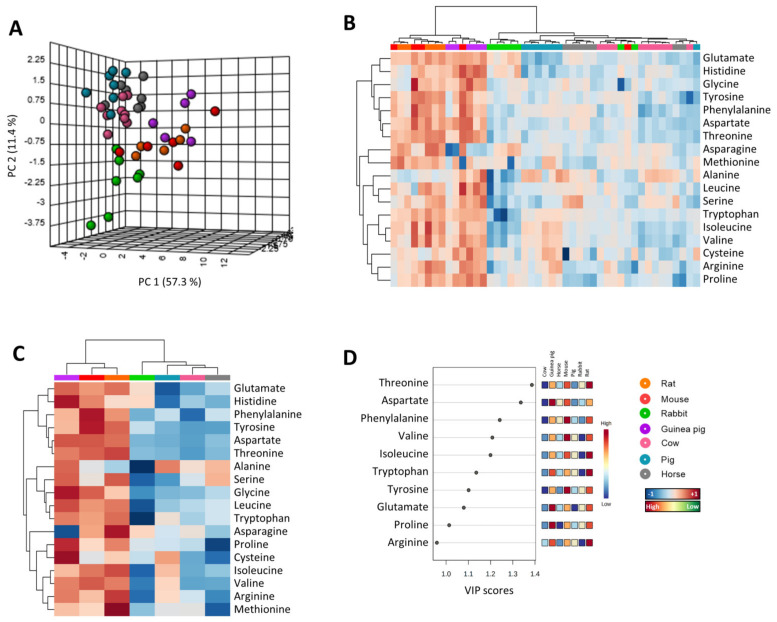
Multivariate statistics reveals a specific amino acids heart profile for mammals. (**A**) Principal component analyses (PCA) representation of amino acids. X: Principal component 1 (PC1); Y: Principal component 2 (PC2); Z: Principal component 3 (PC3). PC3 = 7.7% (not shown). (**B**) Hierarchical clustering of individual animal samples according to amino acids abundance. (**C**) Hierarchical clustering of animal species according to average amino acids abundance. (**D**) Variable importance projection (VIP) scores indicating the elements which contribute most to define the first component of a partial least squares discriminant analysis (PLS-DA).

**Figure 4 biology-11-01821-f004:**
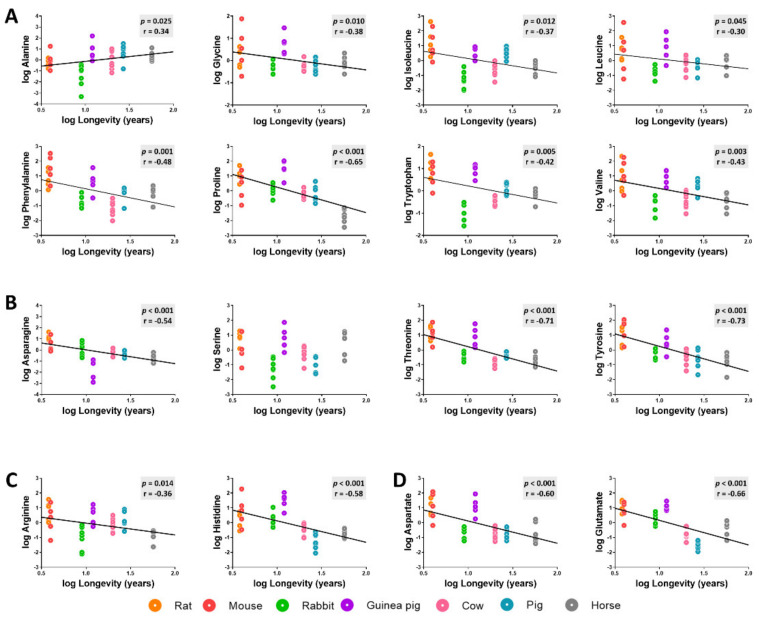
Heart concentrations of non-polar (**A**), polar uncharged (**B**), positively charged (**C**), and negatively charged (**D**) amino acids as a function of mammalian longevity. Individual amino acid heart concentrations are reported in log(mmol/g tissue). Pearson correlation between heart amino acids and animal longevity was performed (**A**–**D**). Linear regression was applied when significant relationships were found. Minimum significance level was set at *p* < 0.05. All amino acid values were log-transformed in order to accomplish the assumptions of normality.

**Figure 5 biology-11-01821-f005:**
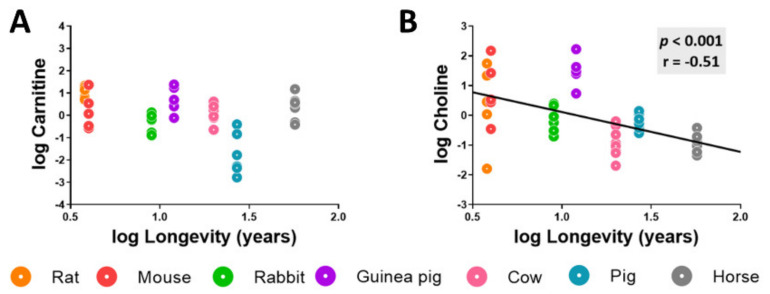
Heart choline concentration negatively correlates with mammalian longevity. Individual heart concentration of the lipid intermediates choline (**A**) and carnitine (**B**) are reported in log(MS counts/g tissue). Pearson correlation between the heart carnitine and choline lipid metabolites and animal longevity was performed. Minimum significance level was set at *p* < 0.05. All values were log-transformed in order to accomplish the assumptions of normality.

**Figure 6 biology-11-01821-f006:**
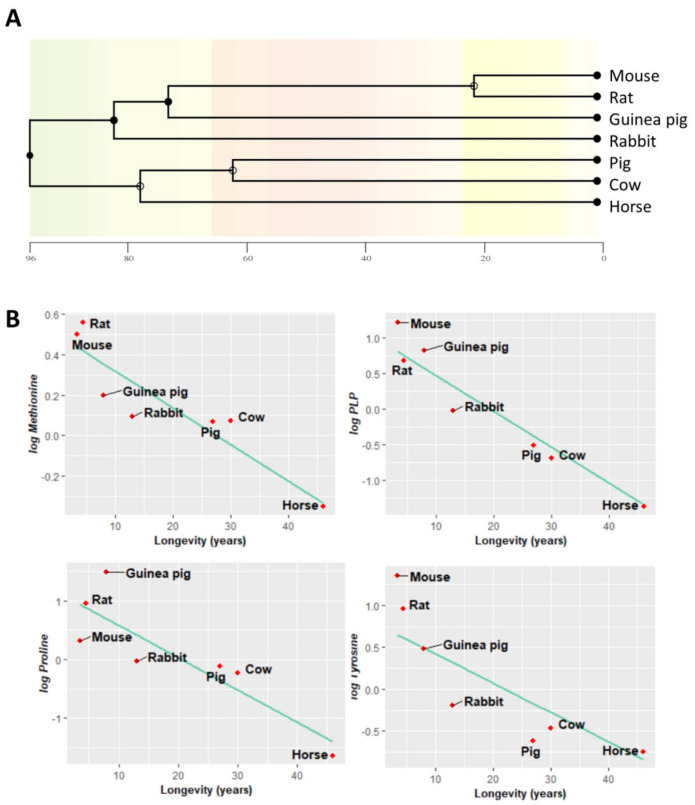
Heart metabolites negatively correlate with mammalian longevity after correcting for phylogenetic relationships. (**A**) Phylogenetic tree of the species studied; the abscissa is expressed in millions of years ago. (**B**) Phylogenetic generalized least squares regression (PGLS) between longevity and the heart concentration of methionine-related metabolites, amino acids, and TCA cycle intermediates. The minimum significance level was set at *p* < 0.05. All metabolites were log-transformed in order to fulfil the assumptions of normality.

## Data Availability

Not applicable.

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
