# Peer review of "Methionine Metabolism Is Down-Regulated in Heart of Long-Lived Mammals"

_biology, 2022, doi:10.3390/biology11121821_

Round 1
Reviewer 1 Report
This is an interesting, well-written and crisp report of the authors’ LC-MS/MS analyses of a series of metabolites related to methionine in several mammalian species of different maximum longevity. The described sampling and analytical protocols appear sound, the quantitative and statistical evaluations as well as the observed effects are robust, but the former may be improved as regards understandability. In my view, this is a very valuable and significant piece of work.
Specific comments:
(1) In the title of their work, the authors summarize their findings by stating: “Methionine metabolism is down-regulated”. However, it is the metabolites that are down-regulated, perhaps by virtue of an upregulated catabolic metabolism. Hence, which alternative is correct?
a – The anabolic systems using and thus removing the metabolites, e.g. protein synthesis (for amino acids) or glycerophospholipid biosynthesis (for choline), are more efficient and have lower apparent Km-values, resulting in lower concentrations of the free metabolites.
b – The metabolites are more rapidly degraded by catabolism, such that a lower intracellular concentration ensues.
c – The import of these metabolites, many of which are provided by the liver through the bloodstream, is less effective. Or, to give it a positive tone, import is less necessary because the components are less rapidly damaged and do not need to be exchanged so often in the heart.
Please discuss.
(2) Are the effects observed for intracellular amino acids (metabolites) stronger, weaker, or similar to the effects in plasma, as reported by the authors in ref. 52? Such a comparison might uncover a potential mechanistic origin of the reported regressions.
(3) What do the components PC1-PC3 in Figure 1 and Figure 3 stand for? Do they have a molecular interpretation?
(4) The unit mM/g does not exist. Maybe mmol/g?
(5) Line 57, since aging is not programmed, the authors may prefer to write “targeting the nuclear anti-aging program” or alike.
Author Response
Reviewer 1
This is an interesting, well-written and crisp report of the authors’ LC-MS/MS analyses of a series of metabolites related to methionine in several mammalian species of different maximum longevity. The described sampling and analytical protocols appear sound, the quantitative and statistical evaluations as well as the observed effects are robust, but the former may be improved as regards understandability. In my view, this is a very valuable and significant piece of work.
R/ We thank the reviewer comment.
Specific comments:
(1) In the title of their work, the authors summarize their findings by stating: “Methionine metabolism is down-regulated”. However, it is the metabolites that are down-regulated, perhaps by virtue of an upregulated catabolic metabolism. Hence, which alternative is correct?
- The anabolic systems using and thus removing the metabolites, e.g. protein synthesis (for amino acids) or glycerophospholipid biosynthesis (for choline), are more efficient and have lower apparent Km-values, resulting in lower concentrations of the free metabolites.
- The metabolites are more rapidly degraded by catabolism, such that a lower intracellular concentration ensues.
- The import of these metabolites, many of which are provided by the liver through the bloodstream, is less effective. Or, to give it a positive tone, import is less necessary because the components are less rapidly damaged and do not need to be exchanged so often in the heart.
Please discuss.
R/ In accordance with the reviewer's comment, we have discussed these comments (lines 456-476)
(2) Are the effects observed for intracellular amino acids (metabolites) stronger, weaker, or similar to the effects in plasma, as reported by the authors in ref. 52? Such a comparison might uncover a potential mechanistic origin of the reported regressions.
R/ After comparing the results:
- Methionine and SAM reduction are similar in heart and plasma (for methionine: -73% in heart and 79% in plasma; for SAM: -62% in heart and -34% in plasma)
- Cystathionine and taurine reduction are stronger in heart than in plasma (for cystathionine: -87% in heart and -72% in plasma; for taurine: -96% in heart and 76% in plasma)
- Asparagine, glutamate and threonine reduction are stronger in heart than in plasma (for asparagine: -91% in heart and -78% in plasma; for glutamate: -66% in heart and -32% in plasma; for threonine: -82% In heart and -56% in plasma)
- Cysteine and glutathione reduction are weaker in heart than in plasma (for cysteine: -55% in heart and -86% in plasma; for glutathione: -16% in heart and 97% in plasma)
- SAH, betaine and choline increases are weaker in heart than in plasma (for SAH: +104% in heart and +921% in plasma; for betaine: +180% in heart and +571% in plasma; for choline: +15% in heart and +165% in plasma)
Globally, changes in plasma are stronger than that of heart, probably due to heart metabolic needs and adaptations. We have included this information in lines 438-455
(3) What do the components PC1-PC3 in Figure 1 and Figure 3 stand for? Do they have a molecular interpretation?
R/ When dealing with multiple variables, principal component analysis (PCA) allows for grouping of multiple variables into a smaller set, named principal components (PC). The aim is to reduce dimensionality without losing information, and PCs are numbered according to the percentage of sample variability capable to explain (PC1 explains more variability than PC2, PC3…). Although the higher the number of PCs used, the higher the variability explained, the objective is to be able to explain the highest sample variability using the lesser number of PCs.
In figure 1, methionine-related metabolites are used to construct a PCA, and PC1 = 42.3%, PC2 = 18.5% and PC3=10.1%. This means that when using one PC (PC1), our data is capable to explain up to 42.3% of variation. When using two PC (PC1+PC2) 42.3+18.5=60.8%, when three PC (PC1+PC2+PC3) = 42.3+18.5+10.1=71.9%. These results tell us that using 3 PC we are capable to explain a high sample variation. The biological interpretation is that methionine related metabolites content allows us to define a specimen’s species, from the ones included in our study. No molecular interpretation from these data can be done. The same applies for figure 3.
Although we believe that our results are promising and valuable, we are aware that our data only evaluates the content of selected metabolites and seven mammals, and the use of a more extended dataset would be useful to evaluate whether these rules apply to other animal species, such as birds.
(4) The unit mM/g does not exist. Maybe mmol/g?
R/ We have corrected the information
(5) Line 57, since aging is not programmed, the authors may prefer to write “targeting the nuclear anti-aging program” or alike.
R/ The evolutionary meaning and basic molecular mechanisms involved in longevity determination remains an unresolved problem. Currently, different theories are proposed to offer a respond to this biological trait and to explain the enormous range of longevities observed in the animal kingdom. These theories can be grouped in those that defence a non-programmed aging, and those propose the existence of a programmed aging. The findings obtained from comparative studies analysing mechanisms associated with animal longevity at different levels of biological organization (from genomics to metabolomics, and from molecular to organismal level) clearly point to the idea that longevity is a programmed biological process. For this reason, we ask the reviewer to allow us to keep the text in its current form.
Reviewer 2 Report
The paper reports on heart tissue concentration in amino acids and some related metabolites in ex vivo samples from different animal species selected to represent different natural longevities. The methods are appropriate and the data are well reported, including tables and figures. The reported data are genuine, new and of scientific interest.
However, the paper also includes some passages that could benefit from a revision.
After reading the methods section several times, I was not able to fully rule out one possible concern on what the reported amino acid concentrations mean: are they the total tissue content including the amino acids contained in the tissue proteins or just the free fraction or a mix of the two? This is extremely relevant because if we are speaking of the whole tissue content, we are comparing the proteome of various species; if rather they are just the free fractions, we are comparing the species metabolic signature. I notice that the initial steps of sample preparation included proteases inhibitors to avoid protein digestion, which may mean that we are dealing with the free fractions, however this passage seems to be not sufficient to completely avoid the mobilization of amino acids from tissue proteins and the data are expressed as mMol per gram of tissue, which seems to indicate a total content. Only in the first sentence of the conclusions the authors openly speak of “free amino acids”. A further clarification of this aspect will benefit the understanding of the readers.
In connection with the above issue, in the conclusions section the Authors mention the possibility that the reduced concentrations of amino acids they reported simply reflect a reduced protein turn over (correct) that would in parallel reduce the fluctuating fraction of the same metabolites. However, the same could happen also in case of reduced protein content. The Authors duly checked the protein content of their samples to calculate the concentrations, thus it is advisable that they share with the readers the actual protein content of the different samples to confirm or exclude the above concern.
Although the Authors state among the limitations of their study that it “does not prove causality between the metabolites and longevity traits”, the whole paper is presented and discussed speaking of “adaptations related to longevity”. This is a serious conceptual mistake. As far as we know, the genetic pressure on evolution does not target longevity. Such pression hits the reproductive process, the only one that is able to deliver new genetic instructions to the species, which cannot be influenced by the longevity. The only way for longevity to impact evolution is the increased consumption of resources from longer living individuals that may negatively affect the reproductive potential of younger individuals. In this respect we may consider the possibility that evolution allowed longer life spans only when longer lasting adults did not impact the resource balance, which is indeed something like “less is better”. However, this does not at all mean that “less improves the life span”, it could easily be the opposite. Several parts of the manuscript should be rephrased to ensure that such concept is crystal clear. For the same reasons, the statement that “human population could benefit from reduced methionine intake” (simple summary), which I could even agree for completely different reasons, is unsustainable on the base of the present data.
Moreover, the evolutionary pressure based on resource consumption also opens an alternative explanation of the present findings, i.e. the reduced amino acid concentration may as well correlate with the adult size of the species, i.e. the larger the final size and its requirement for feed, the more conservative should be the metabolism to avoid a negative reproductive pressure. Such a hypothesis could be tested to exclude this possibility or to include it among the possible explanations.
In the discussion (page 11, line 399 and following) the Authors define methionine as a “redox-reactive residue with the capacity to protect cells from damage by ROS” and explain the reduced methionine as a reduced need for antioxidant protection. This is based on some studies showing that some methionine residues exposed by proteins may oxidize without changing the protein function: it was proposed that this might buffer some oxidative load. Although conceptually acceptable, this mechanism could never be redox significant from a stoichiometric point of view and assuming this as a possible evolutionary pressure is not sustainable. Moreover, these statements out of their original context are potentially misleading. Methionine is not a direct redox agent and as a secondary redox player it has rather an oxidative role: any excess of methionine (including a protein/methionine containing meal) compulsory generates equimolar amounts of homocysteine, which is an oxidant.
In the discussion (page 12, lines 459-461) the Authors wrote: “Since choline is synthetized from betaine, it can be considered as a lipid intermediate that connects methionine and mitochondrial metabolism.” Choline is not synthetized from betaine, it is betaine that is generated from choline by the action of the enzyme Choline Dehydrogenase (CHDH), which is active in mitochondria, mainly in the liver. In addition, in the liver, the main source of choline is tri-methylation of phosphatidyl ethanolamine, which consumes 3 moyeties of SAMe and generates 3 homocysteines (thus consuming 3 methionines). Moreover, it is very likely that phosphatidylcholine (stored into the membranes of hepatocytes) acts as a reservoir of methyl groups that can be mobilized when mitochondria become active and start to feed the re-methylation of homocysteine from betaine (BHMT reaction) using the methyl groups stored as phosphatitdylcholine by the CHDH reaction. Thus, the connection between mitochondria and methionine metabolism is true and very strong, but not based on the process presented by the authors.
In addition, the Authors suffer a potential bias in interpreting the meaning of high or low concentration of metabolites because they did not “measure the metabolic flux”, which applies to the comment to several metabolites in the paper. If we find lower (e.g.) choline amounts, this can be due either to low production because of a limited need or, opposite, to higher consumption due to increased metabolic need. Without a full reconstruction of the metabolic flux simplistic statements should be avoided.
Finally, the Authors should always limit their statements and comments by adding “in the heart” or the like without oversizing their findings as a general rule: This is the picture in the heart, but looking at the same pattern in the liver or kidney we might encounter different or opposite findings and it is advisable, so far, to be conservative.
Author Response
Reviewer 2
The paper reports on heart tissue concentration in amino acids and some related metabolites in ex vivo samples from different animal species selected to represent different natural longevities. The methods are appropriate and the data are well reported, including tables and figures. The reported data are genuine, new and of scientific interest. However, the paper also includes some passages that could benefit from a revision.
R/ We thank the reviewer comment.
After reading the methods section several times, I was not able to fully rule out one possible concern on what the reported amino acid concentrations mean: are they the total tissue content including the amino acids contained in the tissue proteins or just the free fraction or a mix of the two? This is extremely relevant because if we are speaking of the whole tissue content, we are comparing the proteome of various species; if rather they are just the free fractions, we are comparing the species metabolic signature. I notice that the initial steps of sample preparation included proteases inhibitors to avoid protein digestion, which may mean that we are dealing with the free fractions, however this passage seems to be not sufficient to completely avoid the mobilization of amino acids from tissue proteins and the data are expressed as mMol per gram of tissue, which seems to indicate a total content. Only in the first sentence of the conclusions the authors openly speak of “free amino acids”. A further clarification of this aspect will benefit the understanding of the readers.
R/ As the reviewer states, we are dealing with free amino acids. We use mmol/gram of heart tissue to express our data. To clarify this, we have included a sentence in the methods section (lines 195-196).
In connection with the above issue, in the conclusions section the Authors mention the possibility that the reduced concentrations of amino acids they reported simply reflect a reduced protein turn over (correct) that would in parallel reduce the fluctuating fraction of the same metabolites. However, the same could happen also in case of reduced protein content. The Authors duly checked the protein content of their samples to calculate the concentrations, thus it is advisable that they share with the readers the actual protein content of the different samples to confirm or exclude the above concern.
R/ We agree with the reviewer. To clarify this, we have included a sentence in the methods sections (in lines 134-135) stating that protein concentration was equal in all sample prior analysis.
Although the Authors state among the limitations of their study that it “does not prove causality between the metabolites and longevity traits”, the whole paper is presented and discussed speaking of “adaptations related to longevity”. This is a serious conceptual mistake. As far as we know, the genetic pressure on evolution does not target longevity. Such pression hits the reproductive process, the only one that is able to deliver new genetic instructions to the species, which cannot be influenced by the longevity. The only way for longevity to impact evolution is the increased consumption of resources from longer living individuals that may negatively affect the reproductive potential of younger individuals. In this respect we may consider the possibility that evolution allowed longer life spans only when longer lasting adults did not impact the resource balance, which is indeed something like “less is better”. However, this does not at all mean that “less improves the life span”, it could easily be the opposite. Several parts of the manuscript should be rephrased to ensure that such concept is crystal clear. For the same reasons, the statement that “human population could benefit from reduced methionine intake” (simple summary), which I could even agree for completely different reasons, is unsustainable on the base of the present data.
R/ We politely disagree with the reviewer about the idea that amino acids composition in long-lived specimens doesn’t represent an adaptation for the evolution of longevity. Methionine is one of the late amino acids that were fixed in the genetic code (Moosmann, 2021). Although the introduction of amino acids in the genetic code is easy (Liu and Schultz, 2010), changing the composition of an already stable and operative genetic code is difficult. Therefore, the late introduction of Met into the genetic code must have been driven by selective pressures (Jones et al., 2013). Recapping the functions of Met in modern cells allows to discern what drove its fixation into the genetic code. Met is codified for the initiation codon AUG, and thus its necessary for protein synthesis initiation. However, its delayed introduction into the genetic code suggest that it was not essential for protein synthesis in ancient cells (Trifonov, 2000, 2009), suggesting that this was not the main force that drove its fixation. Besides, Met residues are not involved in catalysis reactions (Levine et al., 1996; Lim et al., 2019). Nonetheless, Met introduction temporally overlaps with the origin of aerobic life (Cardona, 2019), thus suggesting that its fixation is essential to allow cell survival under oxygenic conditions.
The idea of “less is better” is used to refer to metabolic and structural adaptations. Compared to short-lived species, long-lived species produce less ROS and build their intracellular structures using oxidative-resistant biomolecules. Since they have less intracellular damage, they don’t need to invest energy in maintaining antioxidant systems, or to replace altered intracellular structures. We believe that the comment made of the reviewer “the only way for longevity to impact evolution is the increased consumption of resources from longer living individuals that may negatively affect the reproductive potential of younger individuals” is compatible with our explanation. The metabolic and structural adaptations lead to less energy consumption, which could be redirected, at some point, to reproductive processes, representing an adaptative evolutionary advantage of long-lived species compared to short-lived ones.
Finally, it has been accepted that calorie restriction promotes health span in humans. Furthermore, in animal models these beneficial effects have been attributed to the selective restriction of Methionine (not carbohydrates, lipids or other amino acids). Evidence of reduced methionine-related metabolites in long-lived specimens, similar to what we have found in long-lived species have also been reported in the discussion. All this evidence, support the idea that humans could benefit from reduced methionine intake. We are aware that studies evaluating the effect of selective reduction of methionine intake remain to be performed. We are also aware of the difficulties of performing such studies: dietary habits and nutrient intake in humans are difficult to be controlled. To clarify this, we have rephrased the sentence in the simple summary.
Moreover, the evolutionary pressure based on resource consumption also opens an alternative explanation of the present findings, i.e. the reduced amino acid concentration may as well correlate with the adult size of the species, i.e. the larger the final size and its requirement for feed, the more conservative should be the metabolism to avoid a negative reproductive pressure. Such a hypothesis could be tested to exclude this possibility or to include it among the possible explanations.
R/ When performing inter-species studies, it has been a matter of controversy whether to correct or not the observed correlations for species body size and/or metabolic rate (Barja, 2014; Speakman, 2005). Although it has been suggested that there is a positive correlation between longevity and body size, up to date, it hasn’t been attributed a physiological mechanism to body size that contributes to the rate of ageing. Besides, there are several species that scape this rule of thumb, such as humans and birds, among others, and live longer than what is expected for their body size (Szekely et al., 2015). Accordingly, many exceptions exist to the Pearl’s rate of living theory of aging (Pearl, 1928) that suggests the existence of an association between species-specific metabolic rate and longevity. Hence, we believe that correcting our data for a correlation that is not shared for all living species, or at least for one of the species included in our analysis, would lead to misleading results. Besides, as stated by Barja (Barja, 2014), correcting physiological associations using mathematical concepts would erase biologically-meaningful information regarding species longevity.
In the discussion (page 11, line 399 and following) the Authors define methionine as a “redox-reactive residue with the capacity to protect cells from damage by ROS” and explain the reduced methionine as a reduced need for antioxidant protection. This is based on some studies showing that some methionine residues exposed by proteins may oxidize without changing the protein function: it was proposed that this might buffer some oxidative load. Although conceptually acceptable, this mechanism could never be redox significant from a stoichiometric point of view and assuming this as a possible evolutionary pressure is not sustainable. Moreover, these statements out of their original context are potentially misleading. Methionine is not a direct redox agent and as a secondary redox player it has rather an oxidative role: any excess of methionine (including a protein/methionine containing meal) compulsory generates equimolar amounts of homocysteine, which is an oxidant.
R/ As stated previously, we defend the idea that Met should have been introduced in the genetic code as a consequence of evolutionary selective pressures. As the reviewer states, Met generates homocysteine. However, the relationship between Met content and homocysteine in physiological processes is not that clear. In fact, we had already observed methionine concentration in plasma doesn’t correlate with plasma concentration of sulphur-derived metabolites in long-lived specimens such as centenarians (Mota-Martorell et al., 2021a) or long-lived mammals (Mota-Martorell et al., 2021b).
In the discussion (page 12, lines 459-461) the Authors wrote: “Since choline is synthetized from betaine, it can be considered as a lipid intermediate that connects methionine and mitochondrial metabolism.” Choline is not synthetized from betaine, it is betaine that is generated from choline by the action of the enzyme Choline Dehydrogenase (CHDH), which is active in mitochondria, mainly in the liver. In addition, in the liver, the main source of choline is tri-methylation of phosphatidyl ethanolamine, which consumes 3 moyeties of SAMe and generates 3 homocysteines (thus consuming 3 methionines). Moreover, it is very likely that phosphatidylcholine (stored into the membranes of hepatocytes) acts as a reservoir of methyl groups that can be mobilized when mitochondria become active and start to feed the re-methylation of homocysteine from betaine (BHMT reaction) using the methyl groups stored as phosphatitdylcholine by the CHDH reaction. Thus, the connection between mitochondria and methionine metabolism is true and very strong, but not based on the process presented by the authors.
R/ We agree with the reviewer and corrected the sentence in the manuscript (line 502-503)
In addition, the Authors suffer a potential bias in interpreting the meaning of high or low concentration of metabolites because they did not “measure the metabolic flux”, which applies to the comment to several metabolites in the paper. If we find lower (e.g.) choline amounts, this can be due either to low production because of a limited need or, opposite, to higher consumption due to increased metabolic need. Without a full reconstruction of the metabolic flux simplistic statements should be avoided.
R/ We agree with the reviewer and corrected the sentence in the manuscript (lines 512-513)
Finally, the Authors should always limit their statements and comments by adding “in the heart” or the like without oversizing their findings as a general rule: This is the picture in the heart, but looking at the same pattern in the liver or kidney we might encounter different or opposite findings and it is advisable, so far, to be conservative.
R/ We agree with the reviewer and have corrected this throughout the discussion.